# Coexisting maternal and child undernutrition in Ethiopia: Spatial modeling and multilevel analysis of consecutive EDHS data

Mekuriaw Nibret Aweke[1]*, Habtamu Abebe Getahun[2], Samuel Teferi Chanie[3], Gelila Yitageasu[4], Gebrie Getu Alemu[2], Asebe Hagos[5], Mengistie Kassahun Tariku[6], Gezahegn Eshetu Mekuriya[1], Habtamu Wagnew Abuhay[2], Lidetu Demoze[4], Gedefaw Abeje[7]

**1** Department of Human Nutrition, Institute of Public Health, College of Medicine and Health Sciences, University of Gondar, Gondar, Ethiopia, **2** Department of Epidemiology and Biostatistics, Institute of Public Health, College of Medicine and Health Sciences, University of Gondar, Gondar, Ethiopia, **3** Department of Physiotherapy, School of Medicine, College of Medicine and Health Sciences, University of Gondar, Gondar, Ethiopia, **4** Department of Environmental and Occupational Health and Safety, Institute of Public Health, College of Medicine and Health Sciences, University of Gondar, Gondar, Ethiopia, **5** Department of Health Systems and Policy, Institute of Public Health, College of Medicine and Health Sciences, University of Gondar, Gondar, Ethiopia, **6** Department of Public Health, College of Health Science, Debre Markos University, Debre Markos, Ethiopia, **7** Department of Nursing, Debre Tabor Health Science College, Gondar, Debre Tabor, Ethiopia

* mekunib@gmail.com

## Abstract

### Introduction

Maternal and child undernutrition remains a major public health challenge globally, with the highest burdens observed in low- and middle-income countries. Coexisting maternal and child undernutrition has serious implications for survival, growth, and quality of life. Maternal and child undernutrition is a complex, multifactorial issue influenced by a web of interconnected determinants including health, socio-economic status, education and environmental conditions.

### Objectives

To examine the spatial distribution and multilevel determinants of coexisting maternal and child undernutrition in Ethiopia using EDHS data from 2000–2016.

### Methods

We analyzed a weighted sample of 33,445 participants from four consecutive EDHS surveys. Spatial autocorrelation, hotspot, and interpolation analyses were conducted using ArcMap 10.8. Multilevel logistic regression was performed in Stata 17. Cluster variability was assessed using ICC, MOR, and PCV. Variables with $p < 0.05$ were considered statistically significant.

**Data availability statement:** The data used in this study are sourced from the Ethiopian Mini Demographic and Health Survey and can be requested from the DHS Program at https://dhsprogram.com/Data/ by following the procedures outlined in the Materials and Methods section of the paper. Access to these data is granted upon registration and approval by the DHS Program.

**Funding:** The author(s) received no specific funding for this work.

**Competing interests:** The authors have declared that no competing interests exist.

**Abbreviations:** AOR: Adjusted Odds Ratio; CI: Confidence Interval; CSA: Central Statistical Agency; EAs: Enumeration Areas; EDHS: Ethiopian Demographic and Health Surveys.

## Result

A total of 33,445 weighted sample was used for this study. The prevalence of coexisting maternal and child undernutrition was 22.87% (95% CI: 22.42, 23.32). Spatial analysis result showed that hot spot areas were concentrated in the northern regions especially Tigray, Amhara, and parts of Benishangul-Gumuz. Multilevel logistics regression analysis result revealed that maternal primary education (AOR = 0.88; 95% CI: 0.78, 0.98), secondary/higher education (AOR = 0.38; 95% CI: 0.29, 0.49), medium household wealth (AOR = 0.85; 95% CI: 0.75, 0.97), high household wealth (AOR = 0.78; 95% CI: 0.69, 0.89), child age 12–23 months (AOR = 2.77; 95% CI: 2.44, 3.16), ANC use (AOR = 0.83; 95% CI: 0.75, 0.92), improved toilet (AOR = 0.83; 95% CI: 0.71, 0.98) and regions.

## Conclusion and recommendations

Coexisting maternal and child undernutrition shows marked geographic clustering in northern Ethiopia. Strengthening maternal education, improving household economic conditions, enhancing ANC use, and expanding sanitation services particularly in high-risk regions are essential to reduce the burden.

## 1. Introduction

Maternal and child undernutrition remains a major current global public health problem with big burdens in low- and middle-income countries by increasing mortality and overall disease prevalence [1]. It encompasses various forms of nutritional deficiencies, including short stature/stunting, wasting, and underweight. In recent decades, the recognition of the coexisting maternal and child undernutrition (CMCU) is increasing [2]. This form of nutritional deficiency reflects the combined effects of poverty, food insecurity, and limited access to healthcare services [3,4].

Globally, approximately 50 million children suffer from wasting, including 16 million who are severely wasted, while an estimated 156 million are affected by stunting [5]. Maternal undernutrition contributes to an estimated 800,000 neonatal deaths each year through small-for-gestational-age births, while stunting, wasting, and micronutrient deficiencies are linked to about 3.1 million child deaths annually. [6]. In low and middle-income countries there is high rates of maternal underweight and shot stature, stunting and wasting affecting millions [7]. It affects 11.39% of mother–child pairs experiencing the triple burden of malnutrition, largely influenced by socio-economic disparities [8]. Maternal and child undernutrition in Ethiopia represents a critical public health challenge, marked by a dual and even triple burden of malnutrition. Maternal undernutrition in Ethiopia affects 32% of pregnant women, while 49.3% of children under five are anemic, 43.1% are stunted, and 27.6% are underweight [9–11].

Coexisting maternal and child undernutrition is a complex, multifactorial issue influenced by a web of interconnected determinants including health, socio-economic status, education, and environmental conditions [12,13]. It is primarily related to the

factors such as low maternal education, early marriage, poor dietary diversity, household food insecurity and lack of ante-natal care [7,14–17]. Poor nutritional knowledge and dietary practices during pregnancy, lack of prenatal dietary advice, and insufficient maternal micronutrient intake are factors aggravating this health problem [11,18,19]. Furthermore, social deprivation, gender and household power imbalances, and inequitable intra-household food allocation exacerbate the problem by limiting maternal access to adequate nutrition and healthcare services [20,21].

Maternal and child undernutrition has profound consequences on health, societal, and economic levels [7,22]. Coexisting maternal and child undernutrition is linked to serious health consequences, impacting their survival, growth, and overall quality of life. Maternal undernutrition increases the risk of low birthweight, aggravating intergenerational cycles of poor health and long-term effects of developmental deficits [23]. Children born to malnourished mothers frequently experience growth faltering, delayed cognitive development, and heightened vulnerability to infections [6]. At the same time, these mothers are at greater risk for adverse pregnancy outcomes, anemia, and reduced physical and economic productivity [24].

In addition maternal undernutrition leads to increased risks of adverse obstetric outcomes including hypertensive disor-ders during pregnancy, antepartum and postpartum hemorrhage, preterm labor, operative delivery, and sepsis/chorioam-nionitis [25–27]. Both maternal and child undernutrition also leads to significant negative health outcomes including poor fetal growth, and increased susceptibility to diseases like diarrhea, pneumonia, and malaria [28–30]. Additional it leads to higher healthcare costs, and impairs cognitive development and educational attainment [31]. Child undernutrition in Ethio-pia was estimated to cost 55.5 billion birr annually, equivalent to 16.5% of the national GDP at the time [32].

Despite several interventions have been implemented to reduce the burdens of undernutrition among women and chil-dren, a significant number of mothers and children in Ethiopia suffered from food insecurity and undernutrition. Coexisting maternal and child undernutrition and its spatial distribution with multilevel determinants was not investigated. Examining the spatial patterns and determinants at different level is essential for identifying high-risk areas and its determinants which will help to context-specific interventions. This study aims to assess the spatial distribution and multilevel determi-nants of CMCU using data from the 2000–2016 EDHS.

## 2. Methods

### 2.1 Study design and study settings

A cross-sectional study design was employed using the Ethiopian Demography and Health Surveys (EDHS) in 2000, 2005, 2011 and 2016 to assess spatial pattern of CMCU in Ethiopia. Ethiopia is located in the horn of Africa. It has a total area of 1,100,000 km2 and lies between latitudes 3° and 15°N, and longitudes 33° and 48°E. Ethiopia is a federal repub-lic comprising twelve ethnically based regional states and two chartered cities Addis Ababa and Dire Dawa. The regional states include Afar, Amhara, Benishangul-Gumuz, Central Ethiopia, Gambela, Harari, Oromia, Sidama, Somali, South Ethiopia, South West Ethiopia Peoples', and Tigray [33].

### 2.2 Study population and data source

The EDHS is a nationally representative survey implemented by the Central Statistical Agency (CSA) in partnership with the Ministry of Health and supported by the United States Agency for International Development (USAID)[34]. They pro-vide publicly accessible, reliable data on maternal and child nutrition, healthcare access, and socioeconomic indicators. The EDHS uses a two-stage stratified cluster sampling design: in the first stage, enumeration areas (EAs) are selected proportionally by region and residence (urban or rural); in the second stage, a fixed number of households is systemat-ically sampled within each selected EA [35]. The source population was all women of childbearing age (15–49 years) in Ethiopia. The study population included mother–child pairs from the 2000, 2005, 2011, and 2016 Ethiopia EDHS where children were under five years of age and resided with their mothers. Only pairs with complete anthropometric data were included, specifically maternal height and child indicators of undernutrition height-for-age (stunting), weight-for-age (under-weight), and weight-for-height (wasting).

## 2.3 Sample size and sampling procedure

This study focused on women aged 15–49 years and their children under five to explore CMCU in Ethiopia. Mothers with children with missing or unrealistic data were excluded, leaving a final sample of over 30,667 pairs (weighted to 33,445) from the four survey rounds (Fig 1).

## 2.4 Study variables

**2.4.1 Dependent variable.** In this study, the dependent variable is CMCU a binary variable coded as 1 when both the mother and child are undernourished, and 0 otherwise. Maternal undernutrition is defined as a height less than 155 cm [12,36]. Maternal height is a reliable indicator of long-term nutritional status and reflects the cumulative effects of childhood undernutrition, which may influence maternal and child health outcomes [33]. While child undernutrition is identified by any anthropometric indicator height-for-age, weight-for-age, or weight-for-height below −2 standard deviations from the WHO growth standards [37].

**2.4.2 Independent variables.** The independent variables in this study are grouped into individual-level and community-level factors.

**Individual-level factors** included in the study were socio-demographic and reproductive health-related factors: age of respondents, education level, marital status, occupation status, sex of household head, sex and age of the child, household wealth status, family size, birth order number, and birth interval. Additional variables included type of birth, number of antenatal visits, place of delivery, contraceptive use, cigarette smoking, current pregnancy status, births in the last five years, type of toilet facility, source of drinking water, and media exposure.

**The community-level** variables included in this study were place of residence, regional classification, community literacy level, community poverty level, and perceived distance to health facilities.

## 2.5 Spatial analysis

**2.5.1 Spatial autocorrelation.** Global Moran's I statistic was calculated using ArcGIS version 10.8 to assess the spatial autocorrelation of CMCU. Moran's I values range from −1 (indicating dispersion) to +1 (indicating clustering), with 0 representing a random spatial pattern [38]. A p-value of less than 0.05 was used to determine statistical significance,

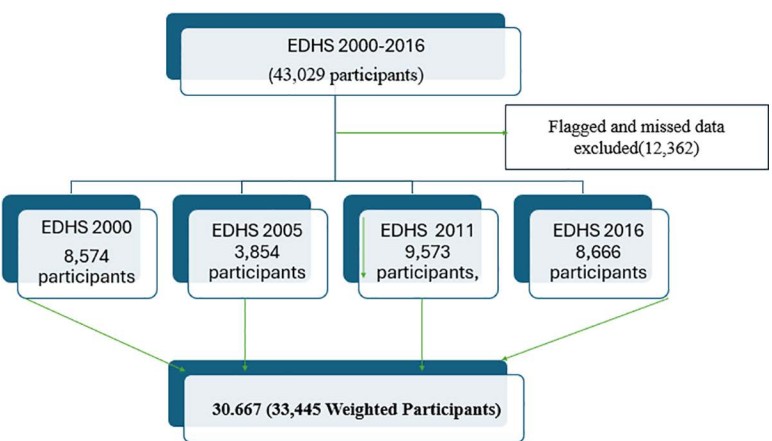

**Fig 1. A schematic representation of the sampling procedure for the study on CMCU among mothers with children aged 0–59 months in Ethiopia, 2000–2016.**

enabling the identification of non-random spatial patterns in CMCU prevalence. The formula to calculate Moran's I is as follows [39,40]:

$$I = \frac{n}{S_0} \frac{\sum_i \sum_j w_{ij} (x_i - \bar{x})(x_j - \bar{x})}{\sum_i (x_i - \bar{x})^2}$$

I = Moran's I statistic

n = number of spatial units (e.g., clusters, regions)

$x_i$ = value of the variable at location i

$\bar{x}$ = mean of the variable

$w_{ij}$ = spatial weight between location i and j

$S_0 = \sum_i \sum_j w_{ij}$ = sum of all spatial weights

**2.5.2 Hot spot analysis.** The Getis-Ord Gi* statistic was applied to conduct hot spot analysis and detect spatial clusters of high and low CMCU prevalence across regions. This method identified statistically significant hot spots areas with higher than expected CMCU and cold spots areas with lower than expected prevalence [41]. This approach helped identify high-risk areas that require more focused attention and resources to combat malnutrition across generations. The Global G statistic can be calculated as follows [40]:

$$G = \frac{\sum_i \sum_{j \neq i} w_{ij} x_i x_j}{\sum_i \sum_{j \neq i} x_i x_j}$$

where,

$x_i$: the value of variable x at location i;

$x_j$: the value of variable x at location j;

$w_{ij}$: the elements of the weight matrix.2.5.3 Spatial interpolation

To estimate CMCU prevalence in unsampled areas, ordinary kriging interpolation was conducted under the assumption that geographically closer locations exhibit more similar CMCU patterns than distant ones. This geostatistical technique enabled the prediction of undernutrition in areas lacking survey data, allowing for a more complete spatial visualization of the burden across Ethiopia [42].

**2.5.4 Spatial scan statistics.** Spatial analyses were conducted using a spatial weights matrix based on inverse distance to account for spatial autocorrelation. Kulldorff's SaTScan version 10.1 software was used to perform spatial scan statistics based on a Bernoulli model, with a maximum spatial cluster size set at 50% of the population at risk, applying a likelihood ratio test with a significance threshold of $p < 0.05$. For spatial interpolation, ordinary kriging was performed using an exponential variogram model to estimate the distribution of CMCU across unsampled locations [43].

## 2.6 Multilevel analysis

A multilevel binary logistic regression in STATA-17 was used to identify factors associated with CMCU among mothers of children aged 0–59 months in Ethiopia. Multicollinearity among independent variables was assessed using variance inflation factors (VIF) and all included variables had VIF values below 10 that indicates no multicollinearity. We applied sampling weights to the DHS data to ensure that the findings are representative of the target population. The analysis considered the hierarchical structure of the data, with individuals and households grouped within enumeration areas (EAs) [44]. A two-level multilevel binary logistic regression model was used, with individuals and households as level one and enumeration areas (EAs) as level two. The model can be expressed as the following [45]:

$$Y_{ij} = \beta_0 + \beta_1 X_{1ij} + \ldots + \beta_m X_{mij} + \varepsilon_{ij} + U_j$$

where $Y_{ij}$ is coexistence of maternal and child undernutrition, $\sigma{ij}$ is the level one variance, $U_j$ is the level two variance, and $m$ is the household-level explanatory variables. Since the outcome variable is binary and the multilevel logistic regression is nonlinear, the above equation can expressed by logit transformation of its probability (*logit* ($P_{ij}$)) as follows [45]:

$$\text{logit } (P_{ij}) = \log\left[\frac{P_{ij}}{1 - p_{ij}}\right] = \beta_0 + \beta_1 X_{1ij} + \ldots + \beta_m X_{mij} + U_j$$

Fixed effects and random effects were included to account for clustering. Four models were fitted: Model I (empty) assessed cluster-level variability; Model II included individual-level variables; Model III included community-level variables; and Model IV included both. Intra-class Correlation Coefficient (ICC), Median Odds Ratio (MOR), and Proportional Change in Variance (PCV) were used to measure between-cluster variation [46].

The Intra-class Correlation Coefficient (ICC) quantifies the proportion of variation in CMCU among women with children aged 0–59 months that is attributable to differences between clusters in Ethiopia [47]. The ICC ranges from 0 to 1, where an ICC of 0 indicates complete independence of observations across clusters. An ICC greater than 0 indicates interdependence of observations within clusters, suggesting that a portion of the variability in CMCU is due to differences between clusters [46]. It is calculated using the formula:

$$\text{ICC } (\rho) = \frac{\sigma^2\varepsilon}{\sigma^2\mu + \sigma^2\varepsilon}$$

where: $\sigma^2\varepsilon$ is the within-group (or residual) variance and $\sigma^2\mu$ is the between-group variance.

The Median Odds Ratio (MOR) quantifies the unexplained cluster-level heterogeneity by expressing the between-cluster variation on the odds ratio scale. It represents the median value of the odds ratio between two individuals with identical characteristics randomly selected from two different clusters one with higher and one with lower risk. The MOR is always greater than or equal to 1; a higher MOR indicates greater between-cluster variability. It is calculated using the formula [46]:

$$MOR = \exp[\sqrt{(2 * VA) * 0.6745}] = \exp(0.95\sqrt{VA})$$

Where; VA is the area level variance, and 0.6745 is the 75th centile of the cumulative distribution function of the normal distribution with mean 0 and variance 1.

The Median Odds Ratio (MOR) is always greater than or equal to 1. If the MOR is 1, it indicates no variation between clusters. The Proportional Change in Variance (PCV) measures the total variation explained by individual and cluster-level factors in each model. It is calculated using the following formula [46]:

PCV = (VA − VB) / VA * 100, where VA = variance of the initial model, and VB = variance of the model with more terms.

## Ethical considerations

Approval for using the EDHS dataset was obtained from the DHS Program/ICF International Inc., and the IRB of the DHS Program https://www.dhsprogram.com. The dataset is de-identified with randomized geographic coordinates to ensure privacy. No additional ethical review was required as there was no direct interaction with participants. We followed all DHS Program ethical guidelines.

## 3. Results

Among the mother–child pairs included in the 2000–2016 EDHS data, most children were male (51.1%) and aged 36–47 months (20.86%). Majority of the mothers were aged 25–34 years (50.8%), had no formal education (72.94%), and were employed (51.43%). The majority were married (91.57%) and resided in rural areas (89.23%). Most households had five or fewer members (57.19%) and two or more children under five (65.85%). Male-headed households were common (86.58%), and nearly half were in the low-income category (45.28%). The majority of births occurred at home (87.44%) via normal delivery (98.77%). Most lived in households with unimproved toilet facilities (88.99%) and non-piped drinking water (84.13%). While 60.94% of mothers had at least one antenatal care visit, only 27.01% of children were vaccinated (Table 1).

### Prevalence of coexisting maternal and child undernutrition in Ethiopia 2000–2016 EDHS

The prevalence of CMCU was 22.87% (95% CI: 22.42, 23.32). The prevalence varied by year and residence. In 2000, it was 27.25%, decreasing to 18.20% in 2016. By residence, the prevalence was 16.46% in urban areas and 23.64% in rural areas (Fig 2).

### Trend of CMCU in Ethiopia (2000–2016)

The weighted prevalence of CMCU in Ethiopia showed a declining trend over the 16-year period. In 2000, the prevalence was 27.2%, which decreased to 23.5% in 2005, 22.7% in 2011, and further declined to 18.2% in 2016. The trend line illustrates a gradual reduction in CMCU across survey years (Fig 3).

### Spatial clustering of coexisting maternal and child undernutrition in Ethiopia

Spatial analysis showed significant clustering of CMCU. Global Moran's I was positive and statistically significant (p<0.001 for 4 round dhs data)suggesting that adjacent regions exhibit similar undernutrition patterns. These high z-scores and low p-values demonstrate that the observed clusters are unlikely to occur by chance (Fig 4).

### Spatial distributions of CMCU in Ethiopia from 2000–2016

The spatial distribution map of CMCU across Ethiopia revealed considerable regional disparities. The highest prevalence was observed in the Amhara region (31.60%), followed by Tigray (26.58%) and Afar (25.60%), whereas the Somali region exhibited the lowest prevalence (5.58%). Central and urban regions such as Addis Ababa (10.52%) and Dire Dawa (14.10%) showed comparatively lower rates(Fig 5).

### Hot spot analysis (Getis-Ord Gi* statistic)

The spatial distribution of CMCU hot spots and cold spots among women and children in Ethiopia using Getis-Ord Gi* statistics reveal significant clustering patterns. Hot spot areas notably concentrated in the northern regions especially Tigray, Amhara, and parts of Benishangul-Gumuz in all consecutive EDHS data from 2000–2016. Conversely, cold spots indicating lower CMCU clustering are mainly located in parts of Gambela, Addis Ababa, Hareri, Dire Dawa and eastern Somali regions (Fig 6).

### Spatial Interpolation

The spatial interpolation results for CMCU in Ethiopia, identifying predicted prevalence across unsampled areas. Lower predicted CMCU burdens concentrated in the eastern, southeastern, and southern parts of the country particularly Somali, parts of Oromia, and SNNPR. In contrast, relatively higher predicted values are observed in regions such as Amhara, Tigray, and central Benshangul Gumz (Fig 7).

**Table 1. *Sociodemographic charachterstics of mothers with children in Ethiopia EDHS 2000–2016.***

| Characteristics | Category | 2000–2016 |
|---|---|---|
| | | N (%) |
| Sex of Child | Male | 17,091(51.10) |
| | Female | 16,354(48.90) |
| Current Age of Child | 0-11 months | 6,814 (20.38) |
| | 12-23 months | 6,517 (19.49) |
| | 24-35 months | 6,458 (19.31) |
| | 36-47 months | 6,978 (20.86) |
| | 48-59 months | 6,678 (19.97) |
| Mother's Age | 15-24 | 7,988 (23.89) |
| | 25-34 | 16,989(50.80) |
| | 35-49 | 8,467 (25.32) |
| Maternal Education Level | No education | 24,394(72.94) |
| | Primary education | 7,326 (21.91) |
| | Secondary and above | 1,724 (5.16) |
| Maternal work status | Unemployed | 16,244(48.57) |
| | Employed | 17,200(51.43) |
| Marital Status | Married | 30,625(91.57) |
| | Unmarried | 2,819 (8.43) |
| Place of Residence | Urban | 3,602 (10.77) |
| | Rural | 29,843(89.23) |
| Number of Household Members | More than 5 members | 14,318(42.81) |
| | 5 or fewer members | 19,127(57.19) |
| Number of Children Under 5 | 2 or more children | 22,023(65.85) |
| | Less than 2 children | 11,421(34.15) |
| Sex of Household Head | Male | 28,955(86.58) |
| | Female | 4,489 (13.42) |
| Wealth Index | Low-income | 10,708(45.28) |
| | Middle-income | 4,951 (20.93) |
| | High-income | 7,992 (33.79) |
| Place of Delivery | Home | 29,245(87.44) |
| | Health Facility | 4,199 (12.56) |
| Mode of Delivery | Normal delivery | 33,023(98.77) |
| | CS delivery | 410(1.23) |
| Type of Toilet Facility | Improved | 3,682 (11.01) |
| | Unimproved | 29,762(88.99) |
| Source of Drinking Water | Non-piped | 28,138 (84.13) |
| | Piped | 5,307 (15.87) |
| Number of Antenatal Visits | No | 13,064 (39.06) |
| | Yes | 20,381(60.94) |
| Birth Order | 1-3 BORD | 16,248 (48.58) |
| | >3 BORD | 17,197 (51.42) |
| Birth type | Single Birth | 32,856(98.2) |
| | Multiple Birth | 589(1.8) |

Footnote. CS: Cesarean Section

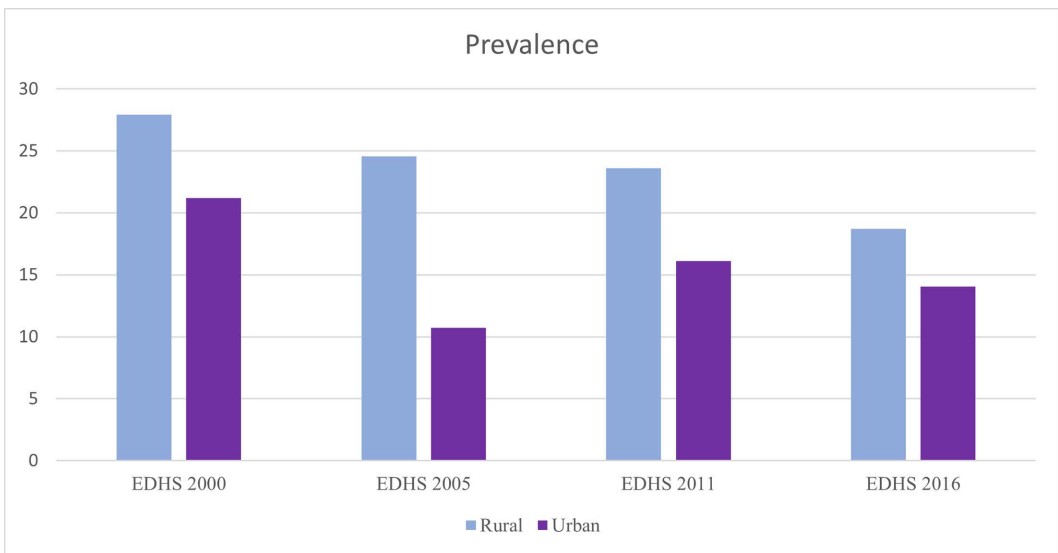

**Fig 2. The prevalence of coexisting maternal and child undernutrition in Ethiopia, 2000–2016.**

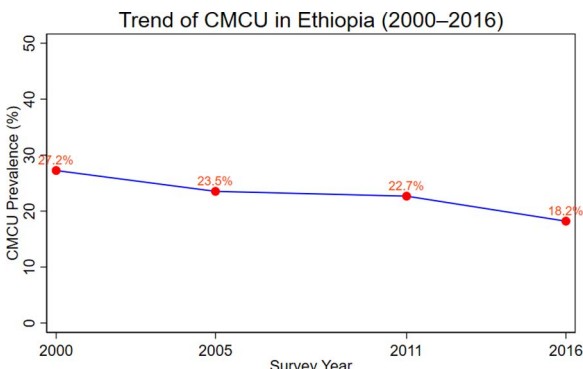

**Fig 3. Trend of coexisting maternal and child undernutrition (CMCU) in Ethiopia, 2000–2016.**

## Spatial clustering of CMCU identified by SaTScan analysis

The results of spatial scan statistics using SaTScan, identifying statistically significant clusters of CMCU in Ethiopia. Each circle represents a cluster, with its size proportional to the population and color indicating the strength of the log-likelihood ratio (LLR). High-risk clusters (in green) are detected in Amhara, Tigray, Hareri and Dire Dawa and SNNPR, where the likelihood of CMCU is significantly higher indicating significantly higher-than-expected cases of CMCU. In contrast, low-risk clusters (in green) are found predominantly in regions such as Addis Ababa, Gambela, and parts of Oromia, is significantly lower (Fig 8).

## Multilevel determinants CMCU in Ethiopia

From the null model, the estimated cluster-level variance of the random intercept was 0.197 (95% CI: 0.161, 0.241), indicating significant variability in CMCU across enumeration areas. As the variance is clearly greater than zero, it confirms the presence of between-cluster differences, supporting the use of multilevel analysis for further investigation. In the

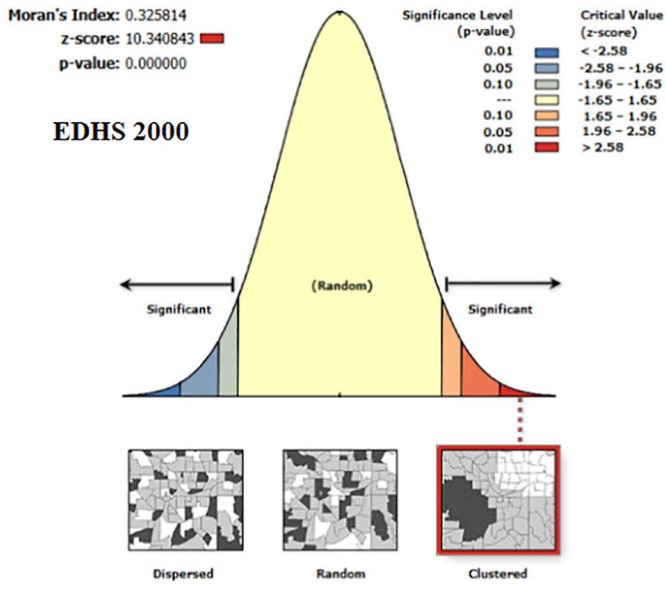

Given the z-score of 10.3408429199, there is a less than 1% likelihood that this clustered pattern could be the result of random chance.

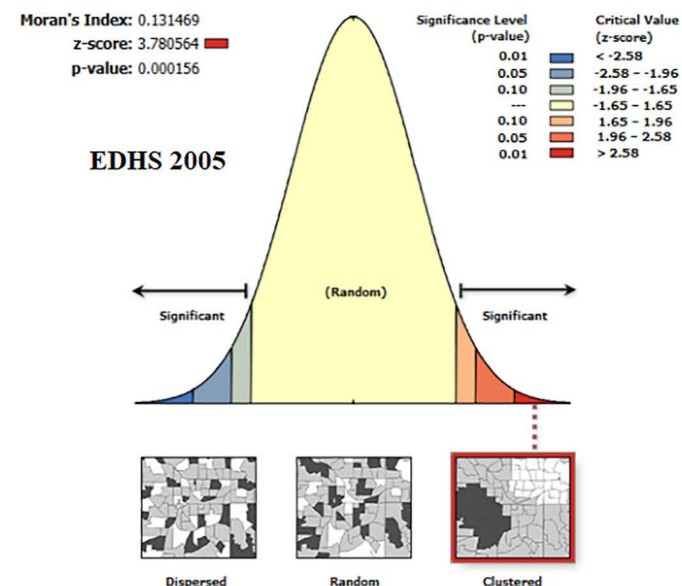

Given the z-score of 3.78056398827, there is a less than 1% likelihood that this clustered pattern could be the result of random chance.

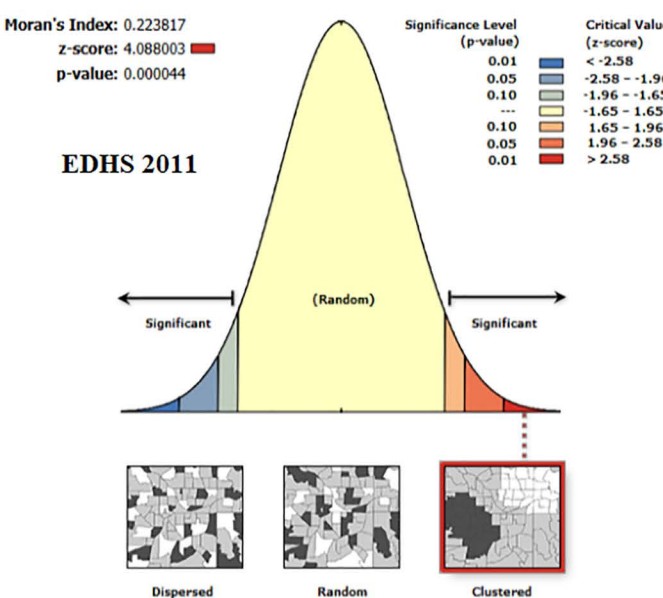

Given the z-score of 4.08800299394, there is a less than 1% likelihood that this clustered pattern could be the result of random chance.

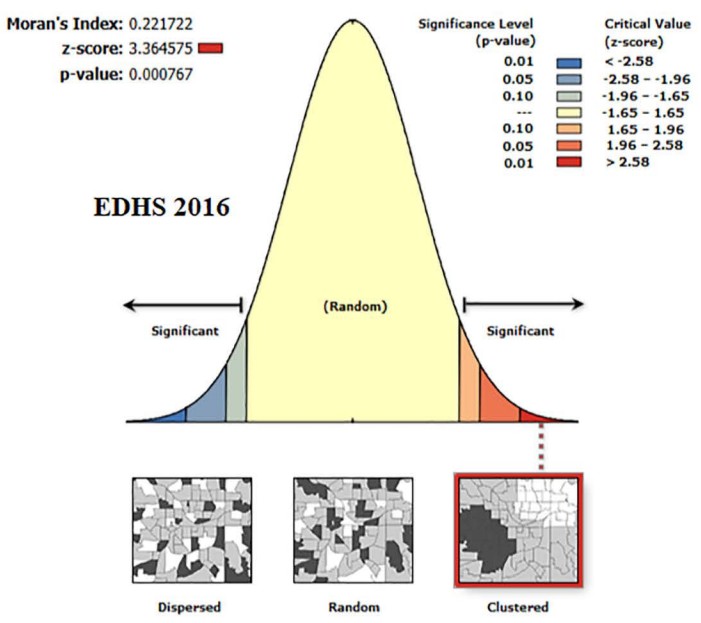

Given the z-score of 3.36457508447, there is a less than 1% likelihood that this clustered pattern could be the result of random chance.

**Fig 4. Spatial clustering of coexisting maternal and child undernutrition in Ethiopia from 2000–2016.**

null model (Model I), the intraclass correlation coefficient (ICC) was 5.64%, indicating that a considerable portion of the variability in CMCU was attributable to differences between clusters. In the full model (Model IV), the ICC decreased to 1.84%, and the proportional change in variance (PCV) reached 68.56%, showing that the included variables explained a

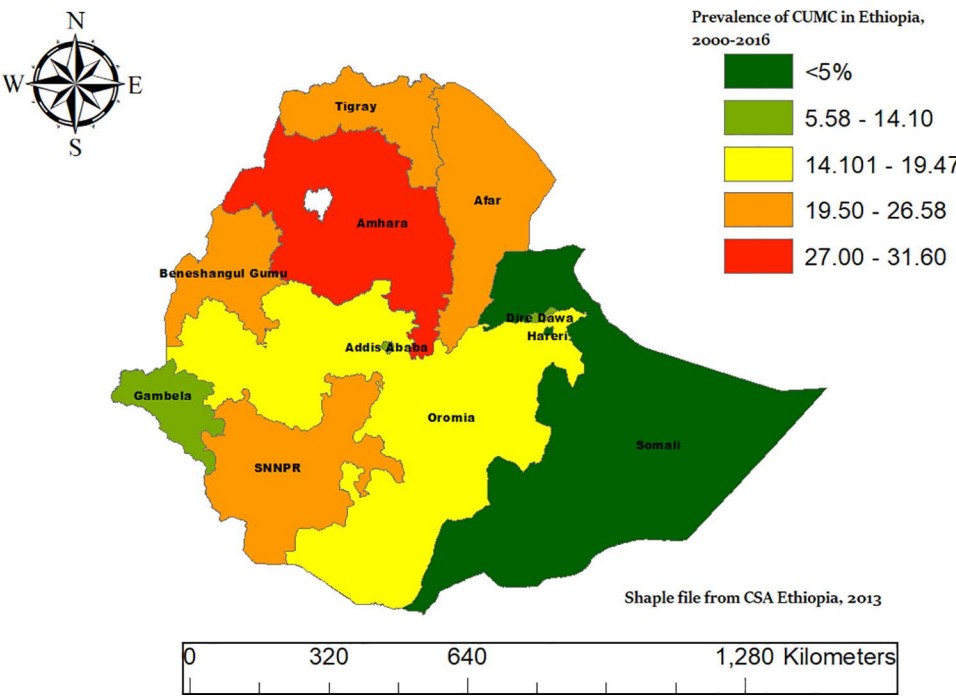

**Fig 5. The prevalence of CMCU across regions in Ethiopia, based on DHS 2000–2016 EDHS data.**

substantial proportion of the between-cluster variation. Additionally, the median odds ratio (MOR) decreased from 1.52 in Model I to 1.27 in Model IV, indicating improved model fit. These results suggest that Model IV provides a better fit to the data by accounting for cluster-level variability (Table 2).

In the multilevel logistic regression analysis, several factors were significantly associated with CMCU among mothers and their children. Maternal education is a significant factor, with children of mothers who had primary education (AOR = 0.88, 95% CI: 0.78, 0.98) and secondary or higher education (AOR = 0.38, 95% CI: 0.29, 0.49) having 12% and 62% lower odds of CMCU, respectively, compared to those whose mothers had no education. Household wealth also had a significant association with children from medium-income (AOR = 0.85, 95% CI: 0.75, 0.97) and high-income households (AOR = 0.78, 95% CI: 0.69, 0.89) were 15% and 22% less likely to experience CMCU compared to those from low-income households. The child's age was strongly associated with CMCU particularly among older children. For instance, children aged 12, 23 months (AOR = 2.77, 95% CI: 2.44, 3.16), 24–35 months (AOR = 2.31, 95% CI: 2.00, 2.68), 36–47 months (AOR = 2.39, 95% CI: 2.04, 2.81), and 48–59 months (AOR = 2.21, 95% CI: 1.84, 2.67) were about 2.2 to 2.8 times more likely to experience CMCU compared to children under one year. Use of antenatal care (ANC) was associated with a 17% reduction in the odds of CMCU (AOR = 0.83, 95% CI: 0.75, 0.92), while access to improved toilet facilities was associated with a 17% lower likelihood of CMCU (AOR = 0.83, 95% CI: 0.71, 0.98). Regional differences were also observed, with significantly lower odds of CMCU in Oromia (AOR = 0.65), Somali (AOR = 0.14), SNNPR (AOR = 0.72), Gambela (AOR = 0.24), Harari (AOR = 0.57), Addis Ababa (AOR = 0.54), and Dire Dawa (AOR = 0.53) compared to Tigray, while children in Amhara had 21% higher odds of CMCU (AOR = 1.21, 95% CI: 1.03, 1.43) (Table 3).

## 4. Discussion

This study the first study aimed to investigates the prevalence of CMCU its spatial distribution and multilevel determinants in Ethiopia using EDHS 2000–2016 data. According to the current study finding the prevalence of CMCU was 22.87%

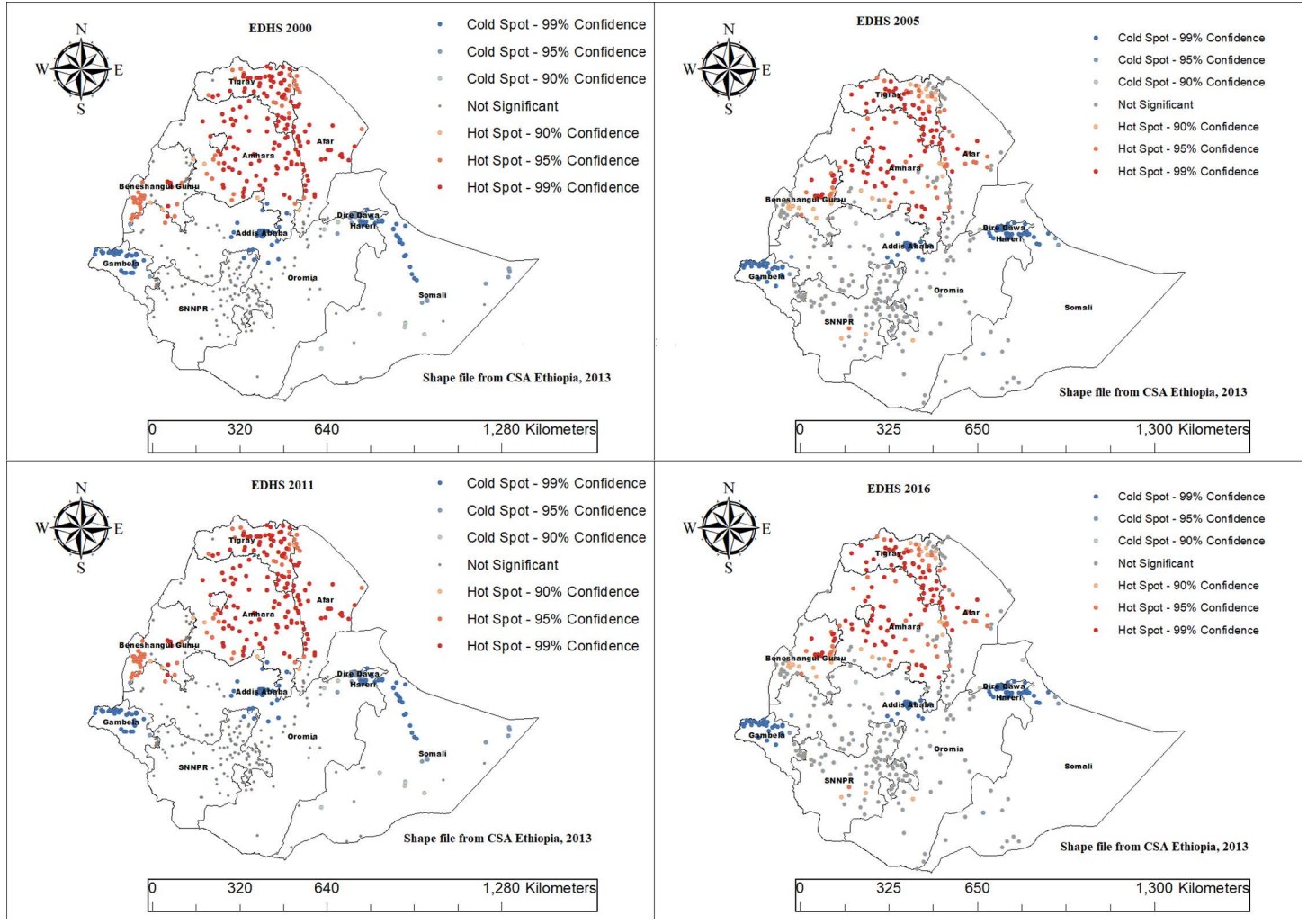

**Fig 6. Hotspot analysis result of CMCU across regions in Ethiopia, based on DHS 2000–2016 EDHS data.**

(95% CI: 22.42, 23.32). Hot spot areas were concentrated in the northern regions especially Tigray, Amhara, and parts of Benishangul-Gumuz.

The spatial clustering of CMCU in the northern regions particularly Tigray, Amhara, and parts of Benishangul-Gumuz suggests that undernutrition is shaped different factors. difference substancial factors such as unique topographical, social, and environmental contribute the hotspots areas found in the Northen parts of Ethiopia. Previous studies on maternal and child undernutrition in Ethiopia have reported similar patterns. The main reason for these is Northern Ethiopia is related with recurrent droughts, soil degradation, and low agricultural productivity [48–50]. These conditions causes for chronic food security and limit dietary diversity and reduce the availability of nutritious foods, thereby increasing the vulnerability of mothers and children to undernutrition. In addition the higher likelihood of CMCU in Amhara and Tigray may also be related to cultural and dietary practices. Previous studies showed that there is low consumption of animal-source foods (ASF) in the northern parts of Ethiopia [51]. Animal-source foods are the good source of proteins and also provide essential nutrients such as iron, zinc and vitamin B12 [52]. All these nutrients are critical for maternal health, fetal growth,

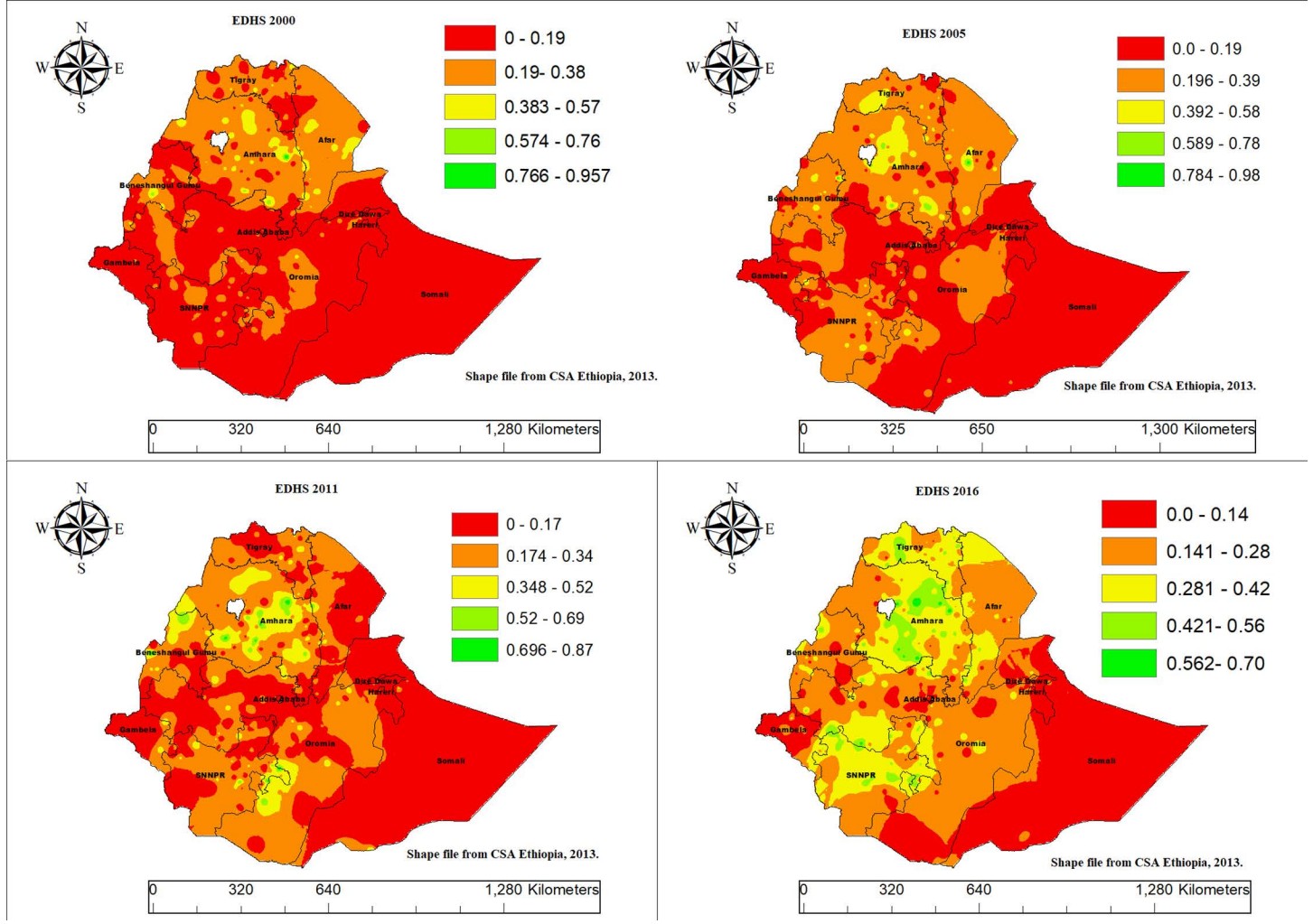

**Fig 7. Spatial interpolation result of CMCU among mother–child pairs across regions in Ethiopia, based on DHS 2000–2016 EDHS data.**

and child development. low consumptions of these foods may therefore contribute to the coexisting of undernutrition observed among mothers and children.

Multilevel logistics regression analysis result showed that maternal education, household wealth, child's age, antenatal care utilization, improved toilet facilities, and region were significantly associated with CMCU. According to the findings of the current study, the multilevel analysis revealed that mothers and their children residing in the Amhara and Tigray regions had a higher likelihood of experiencing CMCU. This could be due to combination of different factors such as socioeconomic, health service, and environmental factors. These regions have faced high levels of food insecurity, limited access to quality healthcare, and poor maternal and child nutrition practices. For instance Tigray and Benshangul regions were often experience recurrent droughts that significantly impact its agricultural productivity [53,54]. Another possible explanation is the particularly low consumption of animal source foods in the northern regions of Ethiopia, notably in Amhara and Tigray [55]. Overall, these three regions are characterized by widespread food insecurity, with a large proportion of households experiencing food shortages with varying degrees of severity [56–59].

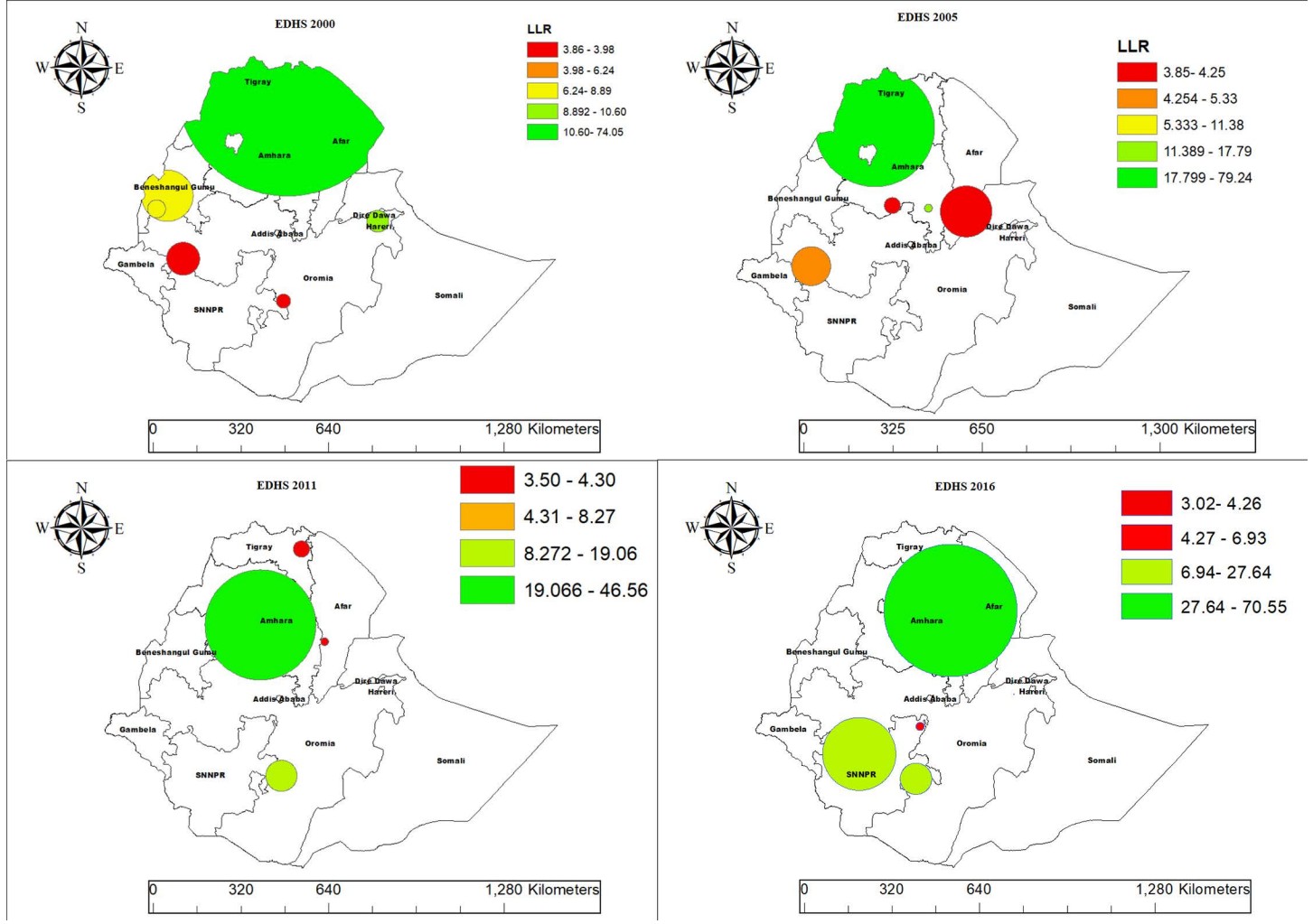

**Fig 8. Spatial interpolation result of CMCU across regions in Ethiopia, based on DHS 2000–2016 EDHS data.**

**Table 2. Measures of variation and model comparison for coexisting maternal and child undernutrition in Ethiopia across four models.**

| Measure of variation | Model – I | Model II | Model-III | Model- IV |
|---|---|---|---|---|
| ICC % | 5.64 | 4.15 | 2.79 | 1.84 |
| PCV % | Reference | 27.56 | 51.87 | 68.56 |
| MOR | 1.52 | 1.43 | 1.34 | 1.27 |
| AIC | 30640.42 | 13773.23 | 29517.14 | 13363.25 |
| Log likelihood | −15318.21 | −6855.613 | −14742.569 | −6636.6275 |
| BIC | 30657.08 | 14009.69 | 29650.43 | 13706.51 |

The current study revealed a strong association between maternal education and CMCU with educated mothers significantly less likely to experience coexisting maternal and child undernutrition compared to those with no formal education. This finding is consistent with previous studies that have shown maternal education significantly reduces the risk of undernutrition [60–62]. The possible explanation for this find could that educated mother are equipped with

**Table 3. Multilevel logistic regression analysis result of coexisting maternal and child undernutrition in Ethiopia 2000–2016.**

| Characteristics | Category | Model I (Empty) | Model II | Model III | Model IV |
|---|---|---|---|---|---|
| Age of Respondents | 15-24 | | 1 | | 1 |
| | 25-34 | | 0.96 (0.85, 1.08) | | 0.98 (0.86, 1.11) |
| | 35,49 | | 0.93 (0.79, 1.09) | | 0.91 (0.78, 1.07) |
| Education Level | No Education | | 1 | | 1 |
| | Primary | | 0.80 (0.72, 0.90)** | | 0.88 (0.78, 0.98)** |
| | Secondary and above | | 0.33 (0.26, 0.43)** | | 0.38 (0.29, 0.49)** |
| Marital Status | Married | | 1 | | 1 |
| | Unmarried | | 1.13 (0.97, 1.32) | | 1.12 (0.95, 1.31) |
| Occupation Status | Unemployed | | 1 | | 1 |
| | Employed | | 0.97 (0.88, 1.07) | | 0.97 (0.88, 1.07) |
| Sex of Household Head | Male | | 1 | | 1 |
| | Female | | 0.83 (0.73, 0.94)** | | 0.89 (0.78, 1.01) |
| Sex of Child | Male | | 1 | | 1 |
| | Female | | 0.95 (0.87, 1.03) | | 0.94 (0.87, 1.03) |
| Child Age | <12months | | 1 | | 1 |
| | 12-23 months | | 2.80 (2.47, 3.19)** | | 2.77 (2.44, 3.16)** |
| | 24-35 months | | 2.38 (2.06, 2.74)** | | 2.32 (2.00, 2.68)** |
| | 36-47 months | | 2.51 (2.14, 2.95)** | | 2.39 (2.04, 2.81)** |
| | 48-59 months | | 2.27 (1.88, 2.73)** | | 2.21 (1.84, 2.67)** |
| Household Wealth Status | Low-income | | 1 | | 1 |
| | Medium-income | | 0.91 (0.81, 1.03) | | 0.85 (0.75, 0.97)** |
| | High-income | | 0.78 (0.69, 0.89)** | | 0.78 (0.69, 0.89)** |
| Family Size | >5 family | | 1 | | 1 |
| | <=5 family | | 1.12 (1.00, 1.25) | | 1.07 (0.96, 1.20) |
| Birth Order Number | <3 BORD | | 1 | | 1 |
| | >3 BORD | | 0.93 (0.82, 1.06) | | 0.94 (0.82, 1.07) |
| Birth Interval (Months) | <36 months | | 1 | | 1 |
| | >=36 months | | 1.02 (0.93, 1.12) | | 0.97 (0.88, 1.06) |
| Type of Birth | Single Birth | | 1 | | 1 |
| | Multiple Birth | | 0.99 (0.66, 1.49) | | 0.98 (0.65, 1.47) |
| Antenatal Visits | No ANC | | 1 | | 1 |
| | Had ANC | | 0.85 (0.77, 0.95)** | | 0.83 (0.75, 0.92)** |
| Place of Delivery | Home | | 1 | | 1 |
| | Health Facility | | 0.89 (0.78, 1.03) | | 0.95 (0.82, 1.09) |
| Contraceptive Use | Do Not Use | | 1 | | 1 |
| | Use | | 1.15 (1.03, 1.29)* | | 1.07 (0.95, 1.19 |
| Cigarette smoking | No | | 1 | | 1 |
| | Yes | | 0.96 (0.59, 1.56) | | 1.08 (0.66, 1.77) |
| Currently pregnant | No | | 1 | | 1 |
| | Yes | | 0.89 (0.77, 1.03) | | 0.93 (0.80, 1.07) |
| Births in Last Five Years | <3 Births | | 1 | | 1 |
| | 3 or More Births | | 0.93 (0.75, 1.15) | | 1.02 (0.82, 1.26) |
| Type of Toilet Facility | Unimproved | | 1 | | 1 |
| | Improved | | 0.72 (0.62, 0.84)** | | 0.83 (0.71, 0.98)** |
| Source of Drinking Water | Non-piped | | 1 | | 1 |
| | Piped | | 0.96 (0.85, 1.09) | | 0.95 (0.84, 1.08) |

*(Continued)*

**Table 3.** (Continued)

| Characteristics | Category | Model I (Empty) | Model II | Model III | Model IV |
|---|---|---|---|---|---|
| Media Exposure | No | | 1 | | 1 |
| | Yes | | 1.08 (0.98, 1.19) | | 1.03 (0.93, 1.13) |
| **Community Level Factors** | | | | | |
| Place of Residence | Urban | | | 1 | 1 |
| | Rural | | | 1.59 (1.42, 1.78)** | 0.91 (0.75, 1.11) |
| Community Illiteracy Level | Low Illiteracy | | | 1 | 1 |
| | High Illiteracy | | | 0.81 (0.71, 0.93)** | 0.89 (0.75, 1.06) |
| Community Poverty Level | Low Poverty | | | 1 | 1 |
| | High Poverty | | | 0.98 (0.88, 1.09) | 0.98 (0.86, 1.11) |
| Community Distance from Health Facility | No Big Problem | | | 1 | 1 |
| | Big Problem | | | 0.96 (0.88, 1.05) | 0.99 (0.89, 1.11) |
| Region | Tigray | | | 1 | 1 |
| | Afar | | | 1.01 (0.89, 1.16) | 1.21 (1.03, 1.43) |
| | Amhara | | | 1.25 (1.11, 1.40)** | 1.21 (1.03, 1.43)* |
| | Oromia | | | 0.65 (0.58, 0.73)** | 0.65 (0.55, 0.77)** |
| | Somali | | | 0.15 (0.12, 0.18)** | 0.14 (0.11, 0.20)** |
| | Benshangul, Gumuz | | | 0.79 (0.69, 0.90)** | 0.89 (0.74, 1.08) |
| | SNNPR | | | 0.80 (0.71, 0.89)** | 0.72 (0.61, 0.86)** |
| | Gambela | | | 0.29 (0.24, 0.35)** | 0.24 (0.19, 0.32)** |
| | Harari | | | 0.46 (0.39, 0.55)** | 0.57 (0.44, 0.73)** |
| | Addis Ababa | | | 0.50 (0.40, 0.62)** | 0.54 (0.39, 0.74)** |
| | Dire Dawa | | | 0.53 (0.45, 0.63)** | 0.53 (0.41, 0.68)** |

good nutritional knowledge, have economical freedom, can access nutrient dense foods and have health care service provision all these reduce undernutrition for both the mother and the child [63–66].

Mothers from wealthy households are less likely to experience CMCU compared to those from poor households. The finding is consistent with previous studies [67–69]. A possible explanation is that wealthier households can afford nutrient-dense foods, including animal-source foods. They are also more likely to basic sanitation facilities [70] to reducing the risk of communicable diseases such as diarrhea, and have better access to healthcare services when illness occurs [71]. These factors collectively contribute to lowering the risk of undernutrition among women and children.

The other significant variable associated with CMCU was child age. When the child gets old the probability of getting CMCU increase. A possible explanation is that as children grow older, their nutritional needs increase, and complementary feeding requires more resources. Without adequate nutrition and care, the risk of illness and undernutrition rises [72]. Additionally, as mothers resume outdoor work activities after childbirth, both mother and child may experience inadequate dietary intake, increasing the risk of undernutrition. Older children also begin crawling and walking, increasing their exposure to contaminated materials and illnesses like diarrhea [73,74]. Moreover, mothers may become pregnant again, further compromising nutritional status due to increased physiological demands.

According to the current study finding antenatal care(ANC) utilization was significantly associated with CMCU compared with those mothers with don't use ANC follow-up. This finding is consistent with previous studies that shows mothers who engage in regular ANC are less likely to experience undernutrition which is crucial for both maternal and child health [75–77]. During ANC follow-up, mothers receive counseling on dietary diversity, healthy lifestyles, and the treatment of infections, including intestinal parasites. These interventions play a significant role in reducing the burden of undernutrition among both mothers and children.

Another significant variable associated with CMCU is the presence of improved toilet facilities in households. Mothers living in households with improved sanitation are less likely to experience CMCU compared to those using unimproved facilities. Previous studies have also demonstrated that mothers and children in households with improved toilet facilities have a lower risk of undernutrition [78,79]. The possible explanation for the association of improved sanitation and reduction of undernutrition among mother and their children could be that access to improved sanitation significantly lowers the risk of nutritional deficiencies primarily by reducing exposure diarrheal diseases, intestinal worm infections, and environmental enteric dysfunction that can impair nutrient absorption [78,80]. Furthermore, Poor sanitation practices increase the risk of infections and physical stress during pregnancy, which can lead to adverse pregnancy outcomes such as low birth weight, preterm birth, and maternal complications [81,82].

## Strength and limitations

This study used nationally representative DHS data. We used a large sample size and standardized methods, ensuring high reliability and generalizability. The application of advanced spatial tools and multilevel modeling provided a comprehensive understanding of both spatial patterns and associated factors. The DHS applies random geographic displacement to cluster coordinates. This can introduce positional error in spatial analyses. This may affect the accuracy of local clustering results including Getis-Ord Gi*, SaTScan and interpolation outputs such as kriging. While global spatial patterns remain largely preserved, localized hotspot locations and boundaries should be interpreted with caution.

## Conclusion and recommendation

This study found that maternal education, household wealth, child's age, antenatal care use, improved toilet facilities, and region were significantly associated with CMCU. High-risk clusters were mainly found in Tigray, Amhara, and parts of Benishangul-Gumuz. We recommend specific interventions in these hotspot regions, with a focus on improving maternal education, household living conditions, and access to health and sanitation services. Addressing these factors through coordinated, multisectoral efforts is essential to break the cycle of undernutrition across generations. Future studies should consider integrating environmental and climatic factors dietary diversity indicators and applying advanced spatial modeling techniques such as GWR or MGWR to better capture spatial variability and improve understanding of the factors contributing to CMCU.

## Author contributions

**Conceptualization:** Mekuriaw Nibret Aweke, Gezahegn Eshetu Mekuriya.

**Data curation:** Mekuriaw Nibret Aweke, Samuel Teferi Chanie, Habtamu Wagnew Abuhay.

**Formal analysis:** Mekuriaw Nibret Aweke, Gebrie Getu Alemu.

**Funding acquisition:** Mengistie Kassahun Tariku.

**Investigation:** Habtamu Abebe Getahun.

**Methodology:** Mekuriaw Nibret Aweke, Habtamu Abebe Getahun, Mengistie Kassahun Tariku.

**Project administration:** Gelila Yitageasu, Asebe Hagos, Gedefaw Abeje.

**Resources:** Asebe Hagos, Gedefaw Abeje.

**Software:** Samuel Teferi Chanie, Habtamu Wagnew Abuhay.

**Supervision:** Mekuriaw Nibret Aweke, Gebrie Getu Alemu, Mengistie Kassahun Tariku, Lidetu Demoze.

**Validation:** Gezahegn Eshetu Mekuriya, Habtamu Wagnew Abuhay.

**Visualization:** Samuel Teferi Chanie.

**Writing – original draft:** Mekuriaw Nibret Aweke, Habtamu Abebe Getahun, Gelila Yitageasu, Gebrie Getu Alemu, Asebe Hagos, Mengistie Kassahun Tariku, Gezahegn Eshetu Mekuriya, Habtamu Wagnew Abuhay, Lidetu Demoze, Gedefaw Abeje.

**Writing – review & editing:** Mekuriaw Nibret Aweke, Habtamu Abebe Getahun, Samuel Teferi Chanie, Gelila Yitageasu, Gebrie Getu Alemu, Asebe Hagos, Mengistie Kassahun Tariku, Gezahegn Eshetu Mekuriya, Habtamu Wagnew Abuhay, Lidetu Demoze, Gedefaw Abeje.

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
