## [Decision Letter · Decision Letter 0]

2 Dec 2025

Dear Dr. Aweke,

Thank you for submitting your manuscript to PLOS ONE. After careful consideration, we feel that it has merit but does not fully meet PLOS ONE’s publication criteria as it currently stands. Therefore, we invite you to submit a revised version of the manuscript that addresses the points raised during the review process.

We look forward to receiving your revised manuscript.

Kind regards,

Daniel Biftu Bekalo, PhD

Academic Editor

PLOS ONE

Journal Requirements:

- https://doi.org/10.3389/fnut.2025.1537348

In your revision ensure you cite all your sources (including your own works), and quote or rephrase any duplicated text outside the methods section. Further consideration is dependent on these concerns being addressed.

3. We note that Figure 4, 5, 6 and 7 in your submission contain [map/satellite] images which may be copyrighted. All PLOS content is published under the Creative Commons Attribution License (CC BY 4.0), which means that the manuscript, images, and Supporting Information files will be freely available online, and any third party is permitted to access, download, copy, distribute, and use these materials in any way, even commercially, with proper attribution. For these reasons, we cannot publish previously copyrighted maps or satellite images created using proprietary data, such as Google software (Google Maps, Street View, and Earth). For more information, see our copyright guidelines: http://journals.plos.org/plosone/s/licenses-and-copyright.

1. You may seek permission from the original copyright holder of Figure 4, 5, 6 and 7 to publish the content specifically under the CC BY 4.0 license.

Reviewers' comments:

Reviewer's Responses to Questions

**Comments to the Author**

1. Is the manuscript technically sound, and do the data support the conclusions?

Reviewer #1: Yes

Reviewer #2: Yes

2. Has the statistical analysis been performed appropriately and rigorously?

Reviewer #1: Yes

Reviewer #2: Yes

3. Have the authors made all data underlying the findings in their manuscript fully available?

Reviewer #1: No

Reviewer #2: No

4. Is the manuscript presented in an intelligible fashion and written in standard English?

Reviewer #1: Yes

Reviewer #2: Yes

Reviewer #1: Review Report of the manuscript titled “Coexisting Maternal and Child Undernutrition in Ethiopia: Spatial Modeling and Multilevel Analysis of Consecutive EDHS Data.”

The reviewer appreciates the authors for their efforts in tackling the important and very pertinent public health problem of concurrent undernutrition in Ethiopian mothers and children. Using geographic modeling, multilevel analysis, and many rounds of EDHS data (2000–2016) offers important insights into long-term trends and regional differences. It is praiseworthy that geospatial methods and hierarchical models have been combined, and this has great potential to guide focused actions and policy development. Overall, the paper addresses a significant issue with a big dataset; nonetheless, a number of significant changes are required to improve methodological rigor and clarity. In order to improve the manuscript's representation in front of a global audience and ensure that it complies with PLOS ONE standards, the authors are asked to make the following major as well as minor changes.

1. The abstract (Lines 27–60) is unnecessarily long and includes repetitive background information. It would be much stronger if it were shortened to the essential elements: the main objective, a brief method summary, the key findings, and a final implication sentence.

2. The introduction (Lines 72–122) contains several overlapping explanations about global undernutrition. These paragraphs should be merged and streamlined so that the central research gap in the Ethiopian context becomes clearer.

3. The spatial analysis component (Lines 185–209) needs more methodological depth. The manuscript should include the formulas and proper citations for Moran’s I and Getis–Ord Gi*, and it should also clarify the type of spatial weights matrix used, the variogram model selected for kriging, and the statistical settings applied in SaTScan.

4. The multilevel modelling section (Lines 210–244) would benefit from additional clarity. Introducing the actual model equation, noting whether DHS sampling weights were incorporated, and briefly explaining how multicollinearity and variable selection were handled would strengthen this part.

5. Figure 3 (Lines ~289–294) is of low resolution and appears pixelated. It should be replaced with a minimum 300-dpi version that has clearer boundaries, a visible scale bar, a north arrow, and a more readable legend.

6. Figure 4 (Lines 295–303) also lacks visual sharpness and color clarity. A higher-resolution version with cleaner symbology and clearer regional labels is needed for publication quality.

7. Figure 5 (Lines 305–312), which presents the hotspot analysis, is blurry. It should be regenerated with higher resolution and should include proper cartographic elements like a scale bar, north arrow, and a more legible legend.

8. Figure 6 (Lines 313–320), the kriging interpolation map, appears pixelated and the color gradient is uneven. A smoother, higher-quality rendering with improved symbology would greatly enhance interpretability.

9. Figure 7 (Lines 321–330), showing the SaTScan clusters, is difficult to read because of overlapping cluster circles. This figure should be redrawn with clearer marker sizes, improved separation, and a refined legend.

10. The reported sample size is inconsistent between the Methods (Lines 38–44) and the Results (Lines 44–50). The manuscript should present one consistent figure throughout.

11. The results section (Lines 270–330) is too descriptive in several places. Presenting the findings in a more succinct, quantitative manner would improve readability.

12. The discussion (Lines 364–425) frequently restates the results instead of interpreting them. This section would be stronger if it focused more on explaining the findings, connecting them to previous research, and examining underlying mechanisms.

13. The conclusion (Lines 426–433) could be enhanced by adding a few explicit directions for future work, such as incorporating environmental variables, dietary diversity indicators, or more advanced spatial modelling approaches like GWR or MGWR.

14. Several typographical and referencing issues appear throughout the manuscript (e.g., Lines 92, 185, 572). These include spelling errors, inconsistent EDHS notation, and the use of non-academic sources like Wikipedia, which should be replaced with authoritative references.

15. The operational definition of CMCU (Lines 165–169) needs stronger justification. In particular, using maternal height under 155 cm as a measure of undernutrition should be supported with literature, and a sensitivity analysis would be advisable since height reflects long-term rather than current nutritional status.

16. Although the dataset spans from 2000 to 2016, temporal patterns are not adequately explored. Including a brief trend analysis or highlighting shifts across survey years would strengthen the interpretation.

17. The manuscript should acknowledge the effect of DHS geographic displacement on spatial accuracy, especially regarding Moran’s I, Gi*, SaTScan, and kriging. This is an important methodological limitation that should be mentioned.

18. The discussion would benefit from incorporating environmental and agro-ecological context. Interpretation of spatial clusters should consider factors such as drought frequency, agricultural productivity, and dominant livelihood systems in each region.

Reviewer #2: This manuscript presents a robust and much-needed analysis of the Coexisting Maternal and Child Undernutrition (CMCU) in Ethiopia using consecutive Demographic and Health Survey (EDHS) data spanning 2000–2016. The study addresses a critical public health challenge, employing advanced analytical techniques.

Study Design and Data: The use of large, nationally representative, and consecutive EDHS data (with a weighted sample of 33,445 mother-child pairs) is a major strength. The cross-sectional design is appropriate for assessing prevalence and associated factors.

Statistical Methods: The analytical approach is appropriate and rigorous. The authors correctly employed multilevel binary logistic regression to account for the hierarchical nature of the EDHS data (individuals nested within Enumeration Areas/clusters), which is essential to produce valid standard errors and risk estimates. The inclusion of standard measures like the Intra-class Correlation Coefficient (ICC), Median Odds Ratio (MOR), and Proportional Change in Variance (PCV) demonstrates a thorough and rigorous approach to model construction and fit assessment.

Spatial Analysis: The range of spatial methods employed (Global Moran’s I, Getis-Ord Gi hot spot analysis, ordinary kriging, and SaTScan) is comprehensive and correctly used to identify geographical clustering and high-risk areas. The finding that CMCU is spatially concentrated in the northern regions (Tigray, Amhara, and parts of Benishangul-Gumuz) is a key finding for targeted intervention.

Conclusions: The conclusions logically follow from the results, identifying maternal education, household wealth, child’s age, antenatal care utilization, improved toilet facilities, and region as significant factors associated with CMCU.

Data Availability Statement

Please make all data available

The authors state, "No—some restrictions will apply," and specify that the data is publicly available through the DHS Program but requires registration and approval.

Recommendation: While the data is publicly accessible via a third-party repository, the need for registration and approval constitutes a technical restriction. Please ensure the manuscript's Data Availability Statement clearly and explicitly states that the data is owned by the DHS Program and is accessible upon request and approval as outlined to fully comply with transparency requirements.

Presentation and Writing Quality

The manuscript is well-structured and intelligible, making the scientific content easy to follow.

Minor issues: I recommend a thorough copyedit to correct minor typographical and grammatical errors. For instance, check for spelling of technical terms (e.g., ensure "Spatial authocorilation" is corrected to "Spatial autocorrelation") and refine a few instances of phrasing for maximum clarity (e.g., in the abstract/objective: "consecutive data the EDHS 2000-2016" )

**Do you want your identity to be public for this peer review?** For information about this choice, including consent withdrawal, please see our Privacy Policy

Reviewer #1: **Yes:** Dr. Arghadeep Bose

Reviewer #2: No

---

## [Author Response · Author response to Decision Letter 1]

8 Dec 2025

08 Dec, 2025

Dear Editors of the Plos One,

We are grateful for the time and expertise devoted to reviewing manuscript PONE-D-25-40555 titled “Coexisting Maternal and Child Undernutrition in Ethiopia: Spatial Modeling and Multilevel Analysis of Consecutive EDHS Data”. We sincerely appreciate the reviewers’ constructive feedback, which has helped us improve the scientific clarity and overall quality of the paper.

In response, we have carefully revised the manuscript and provided a detailed, point-by-point reply outlining how each comment has been addressed. Revisions within the manuscript are clearly marked for easy review. We believe these changes strengthen the analysis, interpretation, and presentation of our findings.

Thank you for the opportunity to revise our work.

Warm regards,

Mekuriaw Nibret Aweke

Corresponding Author

1. Editors comment

1. Please submit your revised manuscript

Authers response:

Dear Editor,

Thank you for the opportunity to revise our manuscript based on the reviewers’ comments and suggestions. We have carefully considered each point and revised the manuscript accordingly. The updated version has now been resubmitted for your review. We hope that these revisions adequately address all concerns and have strengthened the overall quality of the manuscript

Authers response:

Dear editor, thank you very much for the reminder to meets the PLOS ONE’S style requirement. We have revised the manuscript in accordance with the journals style requirement including file naming.

3. We noticed you have some minor occurrence of overlapping text with the following previous publication(s), which needs to be addressed

Authers response:

Dear editor thank you for your reminder to correct some minor text overlap with previous publications. We have revise based on the suggesions and if still there is overlapping of texts we are open to correct accordingly.

4. We note that Figure 4, 5, 6 and 7 in your submission contain [map/satellite] images which may be copyrighted. ---We require you to either (1) present written permission from the copyright holder to publish these figures specifically under the CC BY 4.0 license, or (2) remove the figures from your submission

Authers response:

Dear Editor,

I hope this message finds you well.

Thank you for bringing this to our attention. We confirm that the map images in Figures 4, 5, and 6 were generated using shapefiles obtained from the Central Statistical Agency (CSA) of Ethiopia, hosted on Open Africa Data. These shapefiles, accessible at https://africaopendata.org/dataset/ethiopia-shapefiles , are available under the CC BY 4.0 license, which permits sharing and distribution, including for commercial purposes, with proper attribution. We have updated the figure captions to include the source information and ensure compliance with copyright requirements.

2. Revewer #1 comments

The reviewer appreciates the authors for their efforts in tackling the important and very pertinent public health problem of concurrent undernutrition in Ethiopian mothers and children. Using geographic modeling, multilevel analysis, and many rounds of EDHS data (2000–2016) offers important insights into long-term trends and regional differences. It is praiseworthy that geospatial methods and hierarchical models have been combined, and this has great potential to guide focused actions and policy development.

Response to Reviewer:

Thank you for your positive feedback on the importance of our study. We are very pleased to hear your appreciation, and we value your comments and suggestions, which have helped us improve the manuscript.

1. The abstract (Lines 27–60) is unnecessarily long and includes repetitive background information. It would be much stronger if it were shortened to the essential elements: the main objective, a brief method summary, the key findings, and a final implication sentence.

Response to Reviewer:

Dear Reviewer, Thank you for your helpful comment. We have shortened the abstract by removing repetitive background information and focusing on the key elements objective, brief methods, main findings, and conclusion to make it clearer and more impactful. Thank you again for your valuable comments and suggesions.

2. The introduction (Lines 72–122) contains several overlapping explanations about global undernutrition. These paragraphs should be merged and streamlined so that the central research gap in the Ethiopian context becomes clearer.

Response to Reviewer:

Dear reviewer, thank you for highlighting this point. We have revised the introduction by removing overlapping explanations to clearly present the research gap in Ethiopia. In the revised version, we first describe the overall problem, then outline the global-to-local magnitude and key contributing factors to coexisting maternal and child undernutrition. Finally, we discuss the consequences and emphasize the specific research gap in the Ethiopian context. We appreciate the reviewer’s valuable comment.

3. The spatial analysis component (Lines 185–209) needs more methodological depth. The manuscript should include the formulas and proper citations for Moran’s I and Getis–Ord Gi*, and it should also clarify the type of spatial weights matrix used, the variogram model selected for kriging, and the statistical settings applied in SaTScan.

Response to Reviewer

Dear reviewer we highly appreciate your valuable comment to incorporate the formulas and proper citations for Moran’s I and Getis–Ord Gi*. We had included the formulas with appropraite citation in the revised manuscript and we understand this make the manuscrpt more clear and informative. In addition we have clarified the type of spatial weight matrix used the variogram model selected for kriging and the statistical setting applied in SaTScan. Thank you very much for your invaluable comments.

4. The multilevel modelling section (Lines 210–244) would benefit from additional clarity. Introducing the actual model equation, noting whether DHS sampling weights were incorporated, and briefly explaining how multicollinearity and variable selection were handled would strengthen this part.

Response to Reviewer:

Dear Reviewer, we acknowledge for the important suggestions to include actual model equation, to note about thew weighted sample was utilized for the analysis, and explaining how multicollinearity and variable selection were handled. We have revised the manuscript and all these issues were resolved by including model equation, explaining about sample weight and multicollinearity and handlings of variables.

5. Figure 3 (Lines ~289–294) is of low resolution and appears pixelated. It should be replaced with a minimum 300-dpi version that has clearer boundaries, a visible scale bar, a north arrow, and a more readable legend.

Response to Reviewer:

Thank you for the comment. Figure 3 has been replaced with a high-resolution (300 dpi) version to improve clarity and readability. Since the figure represents a spatial autocorrelation and a scale bar and north arrow are not applicable. Thank you again for your valuable comments.

6. Figure 4 (Lines 295–303) also lacks visual sharpness and color clarity. A higher-resolution version with cleaner symbology and clearer regional labels is needed for publication quality.

Response to Reviewer:

We appreciate your suggesions to increase the quality if figure 4 for publication quality. 7. We have increased the Color clarity and resolution for figure 4 and we hope it is clearly visible.

7. Figure 5 (Lines 305–312), which presents the hotspot analysis, is blurry. It should be regenerated with higher resolution and should include proper cartographic elements like a scale bar, north arrow, and a more legible legend.

Response to Reviewer:

Thank you for your valuable comments regarding the quality of Figure 5, particularly the need for higher resolution and appropriate cartographic elements. We have improved the quality of Figure 5; however, because the figure is based on merged data from five DHS surveys, each originally generated as a separate map, a slight reduction in sharpness occurred during merging. Nonetheless, the figure appears clear and easily readable in the final PDF format used for publication. We hope these improvements address your concerns. The authors remain open to making any further adjustments if additional issues regarding the figure’s quality need correction.

8. Figure 6 (Lines 313–320), the kriging interpolation map, appears pixelated and the color gradient is uneven. A smoother, higher-quality rendering with improved symbology would greatly enhance interpretability.

Response to Reviewer:

Thank you for your valuable comments regarding to figure 6 quality and visibility. We have adress the issue by increasing the resolution and making 300 dpi for better quality of the figure. we hope these improvement adress your concern and we are remaining open to making change if additional correction is needed. Thank you very much for your valuable comments.

9. Figure 7 (Lines 321–330), showing the SaTScan clusters, is difficult to read because of overlapping cluster circles. This figure should be redrawn with clearer marker sizes, improved separation, and a refined legend.

Response to Reviewer:

Dear reviewer, We appreciate the your crucial comments regarding to the difficulty of the figure 7 to read because of overlapping cluster circles.. We have carefully reviewed the figure and made efforts to improve its readability. In the original ArcMap version 10.8 output, there is no overlap of circles; however, when the maps are merged, the visibility reduce by the combined image. Despite this, the figure remains clear and fully interpretable in the final PDF version used for publication. Thank you very much for your valuable comments that will help to enhance the quality of the work.

10. The reported sample size is inconsistent between the Methods (Lines 38–44) and the Results (Lines 44–50). The manuscript should present one consistent figure throughout.

Response to Reviewer:

Thank you very much for bringing to our attention the discrepancy in the sample size reported in the Methods and Results sections of the abstract. The correct weighted sample size is 33,445, but due to a typing error it appeared as 34,445. we have now corrected this in lines 44–50, and it reads 33,445.thank you again.

11. The results section (Lines 270–330) is too descriptive in several places. Presenting the findings in a more succinct, quantitative manner would improve readability.

Response to Reviewer:

Dear reviewer thank you very much fore your suggesions to make the result section more succinct, quantitative manner to improve the readability. we have revised the result section particularly from line 270-330 by making precise, succinct and readable. We appreciate your valuable suggesions.

12. The discussion (Lines 364–425) frequently restates the results instead of interpreting them. This section would be stronger if it focused more on explaining the findings, connecting them to previous research, and examining underlying mechanisms.

Response to Reviewer:

Dear reviewer thank you very much for your crucial comments to integrate the current study finding with previous study finding by connecting them and by justification of the underline mechanisms. According to the comment we have revised the discussion section by clearly explaining the current findings with the relation with thier findings and the underline mechanism for the difference of any similarity. We hope this will adress the reviewer concerns and we remains open to making additional changes if any other comments. Thank you again for the crucial comments and suggesions.

13. The conclusion (Lines 426–433) could be enhanced by adding a few explicit directions for future work, such as incorporating environmental variables, dietary diversity indicators, or more advanced spatial modelling approaches like GWR or MGWR.

Response to Reviewer:

Dear reviewer we appreciate your suggestion to include factors dietary diversity indicators and applying advanced spatial modeling techniques such as GWR or MGWR for future work. According the comments we have revised the conclussion section and reads as the following:

“Future studies should consider integrating environmental and climatic factors, dietary diversity indicators and applying advanced spatial modeling techniques such as GWR or MGWR to better capture spatial variability and improve understanding of the factors contributing to CMCU.”

14. Several typographical and referencing issues appear throughout the manuscript (e.g., Lines 92, 185, 572). These include spelling errors, inconsistent EDHS notation, and the use of non-academic sources like Wikipedia, which should be replaced with authoritative references

Response to Reviewer:

Dear Reviewer, thank you very much for your comments on typographical errors and referencing. We have corrected all identified typographical and spelling issues and replaced the citation that previously relied on a Wikipedia source with authoritative scholarly references. Thank you for your detail review and your help for further improvement of the manuscript.

15. . The operational definition of CMCU (Lines 165–169) needs stronger justification. In particular, using maternal height under 155 cm as a measure of undernutrition should be supported with literature, and a sensitivity analysis would be advisable since height reflects long-term rather than current nutritional status.

Response to Reviewer:

Dear reviewer thank you for your valuable recommendation to support the opretional definition with literatures and to provide strong justification to use maternal hieght to assess maternal under nutrtion. In our study, maternal undernutrition is defined as a height less than 155 cm, a cutoff commonly used in previous studies to identify chronic undernutrition among women of reproductive age (14, 36). Maternal height is a reliable indicator of long-term nutritional status and reflects the cumulative effects of childhood undernutrition, which may influence maternal and child health outcomes. We have revised the oprational definition with strong justification, using literatures to support the cutt of point for height catagorization. We hope these makes the operational definition clearer and more informative. Thank you for your comments again and we remained open to making corrections for additional concerns and comments.

16. Although the dataset spans from 2000 to 2016, temporal patterns are not adequately explored. Including a brief trend analysis or highlighting shifts across survey years would strengthen the interpretation.

Response to Reviewer:

Dear reviewer we thank you for four valuable comment to include temporal trends of the prevalnce across the year 2000 to 2016. Based on your comment to explore temporal patterns, we calculated CMCU prevalence across survey years (2000–2016) and we revised the result section by incorporating this finding. Thank you very much for your help to enhance the quality of this work by including relevant information in the result section.

17. The manuscript should acknowledge the effect of DHS geographic displacement on spatial accuracy, especially regarding Moran’s I, Gi*, SaTScan, and kriging. This is an important methodological limitation that should be mentioned.

Response to Reviewer:

Dear reviewer thank you very much for your recommendations to explain the DHS data geographic displacement and to acknowledge the limitation. Based on your recommendation we have revised the manuscript and the correct stetements are read as the following:

“…The DHS applies random geographic displacement to cluster coordinates. This can introduce positional error in spatial analyses. This may affect the accuracy of local clustering results including Getis-Ord Gi*, SaTScan and interpolation outputs such as kriging. While global spatial patterns remain largely preserved, localized hotspot locations and boundaries should be interpreted with caution”.

18. The discussion would benefit from incorporating environmental and agro-ecologic

---

## [Decision Letter · Decision Letter 1]

13 Jan 2026

Coexisting Maternal and Child Undernutrition in Ethiopia: Spatial Modeling and Multilevel Analysis of Consecutive EDHS Data.

PONE-D-25-40555R1

Dear Dr. Aweke,

We’re pleased to inform you that your manuscript has been judged scientifically suitable for publication and will be formally accepted for publication once it meets all outstanding technical requirements.

Kind regards,

Miquel Vall-llosera Camps

Staff Editor

PLOS One

Reviewers' comments:

Reviewer's Responses to Questions

**Comments to the Author**

Reviewer #1: All comments have been addressed

Reviewer #2: (No Response)

2. Is the manuscript technically sound, and do the data support the conclusions?

Reviewer #1: Yes

Reviewer #2: (No Response)

3. Has the statistical analysis been performed appropriately and rigorously?

Reviewer #1: Yes

Reviewer #2: (No Response)

4. Have the authors made all data underlying the findings in their manuscript fully available?

Reviewer #1: No

Reviewer #2: (No Response)

5. Is the manuscript presented in an intelligible fashion and written in standard English?

Reviewer #1: Yes

Reviewer #2: (No Response)

Reviewer #1: I appreciate the authors' comprehensive and thoughtful responses to all comments made during the review process. The edits significantly improved the manuscript's clarity, methodological rigor, analytical depth, and presentation quality. The abstract and introduction are now shorter and more focused, the spatial and multilevel modelling approaches are clearly justified and openly provided, and the findings and discussion sections are more interpretative and founded in current literature and contextual aspects.

The authors have also appropriately addressed concerns about figure quality, sample size consistency, operational definitions, temporal trends, and methodological restrictions, such as DHS regional displacement. Overall, the work now passes publishing requirements and adds significant value to the literature on mother and child undernutrition from a spatial and multilayered perspective.

I am pleased with the improvements and suggest the manuscript for publication.

Reviewer #2: (No Response)

**Do you want your identity to be public for this peer review?** For information about this choice, including consent withdrawal, please see our Privacy Policy

Reviewer #1: No

Reviewer #2: No

---

## [Editor Report · Acceptance letter]

PONE-D-25-40555R1

PLOS One

Dear Dr. Aweke,

I'm pleased to inform you that your manuscript has been deemed suitable for publication in PLOS One. Congratulations! Your manuscript is now being handed over to our production team.

Kind regards,

on behalf of

Dr. Miquel Vall-llosera Camps

Staff Editor

PLOS One